# Mechanical Properties of a Solvated Biomolecule: RGD (1FUV) Peptide

**DOI:** 10.3390/ijms251810164

**Published:** 2024-09-21

**Authors:** Puja Adhikari, Bahaa Jawad, Wai-Yim Ching

**Affiliations:** 1Department of Physics and Astronomy, University of Missouri-Kansas City, Kansas City, MO 64110, USA; paz67@umkc.edu (P.A.); bahaa.a.jawad@uotechnology.edu.iq (B.J.); 2Department of Applied Sciences, University of Technology, Baghdad 10066, Iraq

**Keywords:** RGD peptide, mechanical properties, solvated biomolecule, density functional theory

## Abstract

The mechanical properties of proteins/peptides play an essential role in their functionalities and implications, as well as their structure and dynamic properties. Understanding mechanical properties is pivotal to our knowledge of protein folding and the molecular basis of diverse cellular processes. Herein, we present a computational approach using ab initio quantum mechanical calculations to determine the mechanical properties—such as bulk modulus, shear modulus, Young’s modulus, and Poisson’s ratio—of a solvated Arg-Gly-Asp (RGD) peptide model. Since this peptide serves as the RGD-directed integrin recognition site and may participate in cellular adhesion, it is considered a promising small peptide for medicinal applications. This successful approach paves the way for investigating larger and more complex biomolecules.

## 1. Introduction

Understanding the molecular structure of peptides, proteins, and other biological molecules is pivotal for discerning their function. The structure is primarily determined by experimental means. Historically, biomolecules have often been studied in crystalline form, a process offering advantages such as enabling high-resolution structural analysis using techniques like X-ray crystallography [1]. While this approach yields valuable insights, biomolecules within crystalline matrices are frequently constrained and may not exhibit the behaviors representative of natural environments.

Studying biomolecules in their native environments, such as in aqueous solutions, is essential for achieving a comprehensive understanding. Analyzing biomolecular structures in solvated form is crucial, given that all known biological processes within the human body occur in aqueous environments. Various in-solution techniques, including NMR spectroscopy, small-angle scattering (SAS), circular dichroism (CD), and infrared (IR) spectroscopy, facilitate such investigations [1]. Water, with its unique properties, plays a pivotal role in the folding processes of biomolecules [2,3].

Large-scale computational modeling of biomolecules in solution holds particular significance as they enable the simulation of biomolecular behavior to accommodate the more realistic environments. This approach allows for a deeper understanding of biomolecular dynamics and interactions, giving insights at the atomistic level, bridging the gap between experimental observations and biological reality [4].

In conventional materials science, we already possess a diverse set of techniques that enable us to gauge the behavior of its structures that can be applied to biomolecules. However, to fully understand the function of biomolecules, it is essential to identify the properties—structural, dynamical, and mechanical—that are critical for design [5]. Proteins, being soft in nature, can easily change their biological function when subjected to mechanical deformation [6]. Many cellular processes involve mechanical forces or deformation at various levels, including cellular, subcellular, and molecular scales [7,8]. For instance, biomolecular motors and machines transform chemical energy into mechanical motion to perform a wide range of functions [9,10]. During cell migration, the cell must generate contractile forces to move forward [11]. The adhesion of cells to extracellular matrix (ECM) via focal adhesion complexes is influenced by the substrate’s stiffness [12,13]. Additionally, all living cells are constantly subjected to gravitational and other forces, with both normal and diseased conditions being dependent on or regulated by their mechanical environment. Such deformation can explain the molecular basis for many of the cellular processes involving mechano-sensing and mechano-transduction [6].

In molecular biology, despite the abundance of data on tumors, the fundamental differences between malignant and benign tumors remain unclear. There are studies that focus on whether the mechanical differences between normal and cancer cells can be used to diagnose cancer progression [14]. In fact, research has suggested that elastic properties such as Poison’s ratio and Young’s modulus of cancerous skin could become the first step in turning elasticity into a clinical tool [15,16].

In this study, we employed the ab initio method to investigate the mechanical properties of a solvated Arg-Gly-Asp (RGD) peptide (PDB ID: 1FUV). The RGD (1FUV) peptide consists of only 11 amino acid (AA) residues (see Figure 1) and is composed of AA sequence Arg-Gly-Asp (RGD) [17]. RGD has a high affinity for the membrane protein called integrin and is used to target cancer cells [18]. In addition, RGD has numerous applications in biomaterial design and biomedical devices. It is used for wound healing, radiotracers in imaging, and implantable medical devices [19,20]. RGD (1FUV) is of appropriate size for analysis of its mechanical properties. To study the mechanical properties, we apply ten different strain percentages to the solvated 1FUV. The details of the methodology are discussed in Section 5, models and methods. This study serves as an initial step in comprehending supersoft biomolecules from various angles.

## 2. Results

### 2.1. Mechanical Properties

In this study, we have computed the bulk modulus (*K*), shear modulus (*G*), Young’s modulus (*E*), and Poisson’s ratio (*η*) of solvated RGD (1FUV). The specifics of the computational methods employed to calculate the mechanical properties are explained in Section 5, models and methods. Based on our recent findings in the amino acid study [21], we noted that mechanical properties exhibit greater variabilities at lower strain percentages. As a result, we have selected ten strains—0.25%, 0.50%, 0.75%, 1.00%, 1.25%, 1.5%, 1.75%, 2.00%, 2.25%, and 2.50%—for evaluating these mechanical properties. The results are listed in Appendix A. Given the fact that the solvated biomolecules may contain voids, which could introduce uncertainty into both of their structure and properties, we performed five sets of calculations for each strain. These five sets of data are utilized for statistical error analysis (see Appendix A and Figure 2) employing standard deviation. In Appendix A, *K1* to *K5* are five sets of calculated bulk moduli, *G1* to *G5* are five sets of calculated shear moduli, *E1* to *E5* are five sets of Young’s moduli, and *η1* to *η5* are five sets of Poisson’s ratios.

The mean values and error bars for the five datasets in Appendix A for *K*, *G*, *E*, and *η* are shown in Figure 2a–d. *K*, *G*, and *E* exhibit a decrease with the increase in the strain percentage, with minor fluctuations as strain rises. *η* also decreases with some fluctuations as the strain increases, but the overall decrease slope is slightly lower than that of *K*, *G*, and *E.* Apparently mechanical properties of solvated RGD (1FUV) follow the trend in solvated AAs [21]. Notably, *E*, *G*, and *η* exhibit relatively higher error at 0.25%, whereas *K* exhibits higher error at 1.75%. These errors can be deemed acceptable.

At 0.25% strain, *K*, *E,* and *η* exhibit the highest mean values of 6.58 ± 0.09 GPa, 9.08 ± 0.21 GPa, and 0.270 ± 0.005, respectively. At 0.50% strain, G has the highest mean value of 3.58 ± 0.02 GPa. Conversely, at 2.25% strain, *K*, *G*, and *E* display the lowest mean values of 5.86 ± 0.10 GPa, 3.42 ± 0.02 GPa, and 8.58 ± 0.06 GPa, respectively. At 1.75% strain, *η* exhibits the lowest mean value of 0.255 ± 0.003. This shows that at lower strain the mechanical properties are slightly higher than the ones with higher strain. In comparison with amino acids [21], the overall *K*, *E,* and *η* exhibit higher values, whereas *G* shows similar values.

Both Poisson’s ratio and Young’s modulus were examined in a cancerous tissue, as these properties of cancerous skin could serve as the first step in using elasticity as a clinical tool [15]. They can also supplement as an additional non-invasive diagnostic tool [15]. The range of the mean values of the *η* of solvated RGD (1FUV) is from 0.255 ± 0.0001 to 0.270 ± 0.005. An interesting point is that the Poisson’s ratio for cancerous skin is about 0.43 [15]. RGD is also known to target cancer cells. At this moment, we are not sure how these values can be correlated. Another point of interest is the stiffness of biomolecules. Stiffness is a mechanical property demonstrated by Young’s modulus. There are studies that show the relationship between stiffness and invasion of cancer cells [16]. During cancer progression, cells can become either soft or stiff [16]. However, there is no firm conclusion yet about the invasiveness and the softening of different types of cancers [14,22]. We trust our pioneering result facilitates the understanding of the elasticity of RGD (1FUV) that helps to target cancer cells.

### 2.2. Bonding Analysis

Bond order (BO) serves as a measure of bond strength. Within the OLCAO scheme, we can compute BO for all bonds. The details of BO are explained in Section 5, models and methods. In the current work, we have applied the cutoff of 4.5 Å for bond lengths (BL). Moreover, we can determine the precise BL and BO for all atomic interactions in the solvated RGD (1FUV). By summing up all BOs within the system and normalizing by the volume containing these bonds, we can calculate the total bond order density (TBOD), a concept used in materials science. Calculating TBOD for biomolecules has its limitations, as biomolecules may not be as compact as crystalline or amorphous materials. However, this concept can still provide a rough analysis. TBOD of solvated RGD (1FUV) stands at 0.019 e^−^/Å^3^. However, due to inherent limitations in applying TBOD for supersoft solvated biomolecules, it requires caution for relative comparisons. Below is a list of materials along with their respective TBOD values just for reference: pyrophosphate crystal K_2_Mg(H_2_P_2_O_7_)_2_ H_2_O (0.021 e^−^/Å^3^) [23], C-S-H cement mineral suolunite (0.027 e^−^/Å^3^) [24], amorphous-SiO_2_ (a-SiO_2_ glasses) (0.025 e^−^/Å^3^) [25], crystalline montmorillonite clay (0.033 e^−^/Å^3^) [26], solvated montmorillonite clay (0.022 e^−^/Å^3^) [26]. Notably, these systems exhibit slightly higher TBOD compared to solvated RGD (1FUV), implying less internal cohesion in the solvated RGD.

It is important to recognize that direct comparisons between biomolecules have not been conducted, rendering such assessments inconclusive. Despite this, drawing from our experience, we maintain that solvated RGD (1FUV) retains considerable strength. This strength is bolstered by its bimolecular composition, wherein the presence of water molecules facilitates the formation of a robust hydrogen bonding network, further fortifying its structural integrity.

Additionally, we can precisely determine the bond strength for interactions between AAs, intra-AAs, AAs with water, inter-water molecules, and intra-water molecules. It is noteworthy that intra-AAs and intra-water molecules typically exhibit higher bond strengths compared to inter-AAs, inter-water molecules, and interactions between AAs and water. Nevertheless, these inter-interactions remain significant and are discussed below in Section 2.2.1 and Section 2.2.2.

#### 2.2.1. Inter-Bonding between Amino Acids

The amino acid—amino acid bond pair (AABP) [27] is an important parameter to study the overall interactions between amino acid residues which may be crucial to determining the conformation and structure properties of this small peptide. AABP provides an overview of twists and turns in peptides or proteins. The details of AABP are explained in the models and method section. In Appendix A, we show AABP values for these 11 amino acid residues of RGD (1FUV). The TAABP can be further divided into contributions from NN and NL interactions. We can further quantify contribution from overall hydrogen bonding (HB), and subtracting it with TAABP, we can obtain contribution from remaining bonding. We also plotted the AABP contributed from nearest neighbor (NN) and non-local (NL) AAs separately in Figure 3a and contribution from hydrogen bonding and other bonding in Figure 3b.

Below are some points observed from Appendix A.

Arg5 has the highest TAABP value, and its contribution is mainly from NN. In fact, Arg5 has the highest contribution from NN, NL, and HB. Coincidentally, Arg is one of the AA residues in the tripeptide RGD. In 1FUV the RGD are Arg5-Gly6-Asp7.

Gly11 has the lowest TAABP value, which is normal since Gly11 only has one NN AA residue and one NL AA residue interaction. In addition, Gly11 has the lowest contribution from HB.

Ala1 has the lowest contribution from NN, followed by Gly11. They are the first and last AA residue in the sequence and have one NN interaction each.

Asp4 has the lowest contribution from NL interaction, as it has only one NL AA residue.

Cys8 has a maximum number of NL AA residue interactions.

Figure 3b shows that Arg5 has the highest TAABP and the highest contribution from HBs, while Gly11, with the lowest TAABP, also has the lowest contribution from HBs. As previously mentioned, Arg5 boasts the highest TAABP, a crucial factor within the context of AA from the tripeptide RGD. The RGD sequence serves as a crucial cell attachment site for various adhesive proteins found in the extracellular matrix, blood, and cell surfaces. Interestingly, more than 20 known integrins recognize this sequence within their adhesion protein ligands [28]. It is noteworthy that each of Arg5-Gly6-Asp7 exhibits two NL AA residue interactions, enabling them to actively participate in cell adhesion.

#### 2.2.2. Inter-Bonding between Amino Acids and Water Molecules

Appendix A and Figure 4 focus on bonding between AA residues of RGD (1FUV) peptide and H_2_O. This analysis delves into identifying the HB within these interactions, as detailed in Appendix A. Figure 4 illustrates that the contribution from HB is comparatively higher than other bonding.

From Appendix A, it can be noticed that Cys2 has the lowest BO when interacting with H_2_O, no HB interaction. Following Cys2, Cys8 has the second lowest BO when interacting with H_2_O. The reasoning is the hydrophobic nature of Cys residue, and they usually isolate from the polar solvent. Coincidently, both Cys2 and Cys8 participate in disulfide bonds with Cys10 and Cys4, respectively, and have lower interactions with water. Cys residues are involved in three-dimensional structure stabilization through the formation of disulfide bridges. Following Cys2 and Cys8, Phe9 also has lower BO when interacting with H_2_O due to its hydrophobic nature. On the other hand, Asp3 has the highest interactions with H_2_O due to its hydrophilic nature. This analysis above gives us an overview of the impact of amino acid residue interactions at the atomistic level for the first time. However, to establish further correlations with significant implications, additional studies of this type are required.

## 3. Discussion

In this study, we have calculated the mechanical properties of solvated RGD (1FUV), creating a pathway of unprecedented analysis for large biomolecules. In addition, we have conducted detailed bonding analysis, including TBOD. Based on our analysis, we conclude that RGD (1FUV) is a reasonably cohesive peptide. We plan to make further ambitious connections between the mechanical properties of ultrasoft biomolecules and bonding analysis based on ab initio quantum chemical calculations.

The mechanical properties of materials are typically rather straightforward, such that a single value can represent their overall properties. However, the scenario is significantly different when considering biomolecular entities such as peptides and proteins. In biomolecules, certain regions may exhibit more flexibility than others due to the varied permutations and combinations of multiple amino acid residues. The identification of flexible and rigid regions within peptides and proteins is crucial for understanding the mechanism of protein folding [29,30]. In addition, since there is ongoing research on the mechanical properties of cancer cells, our results on the mechanical properties of RGD, which is used to target cancer cells, may provide valuable insights in this area. While we have conducted the first study to calculate the mechanical properties of a peptide, much work remains to be done. We plan to further analyze the mechanical properties of other biomolecules by leveraging the mechanical properties of amino acids. However, such analysis requires a substantial database, which is time-consuming. We plan to establish further connections with amino acids to ascertain if we can predict these values solely based on the types and specific quantities of amino acid residues present in biomolecules. We would like to emphasize that this study is foundational, and it often takes years of fundamental research before such findings become applicable. Based on the AABP study, we can quantify the bonding strength of each involved amino acid residue. Additionally, we can quantify the contribution from HB and the number of NL AA residue interactions. As such, we identified that each of Arg5-Gly6-Asp7 exhibits two NL AA residue interactions, enabling them to actively participate in cell adhesion. This could be the reason behind RGD’s high affinity for integrin and its use in targeting cancer cells.

## 4. Conclusions

In this study, we have, for the first time, calculated mechanical properties of a solvated RGD peptide using ab initio quantum mechanical calculations. Additionally, we performed bonding analysis, including AABP and TBOD, highlighting RGD’s cohesion and its relevance in cancer research due to its affinity for integrin. Our future work will extend this approach to other biomolecules, aiming to develop predictive models based on amino acid residue composition.

## 5. Models and Methods

The structure of RGD (1FUV) is obtained from the RCSB protein data bank (PDB) [31,32], which contains 19 models with identical compositions and numbers of atoms. We selected the first model, which consists of 11 amino acid residues with a total of 135 atoms. The peptide is then solvated using PACKMOL software v20.15.1 [33] with the solvation shell of 3 Å and adding 155 water molecules. The RGD (1FUV) with added water molecules has a total of 600 atoms, as shown in Figure 1b. This solvated RGD (1FUV) model is optimized via the Vienna ab initio simulation package (VASP). In VASP, we used the projector augmented wave (PAW) [34,35] method of Perdew–Burke–Ernzerhof (PBE) [34], one of the best exchange correlation functionals within the generalized gradient approximation (GGA). We used the following input parameters: energy cutoff of 600 eV, electronic convergence of 10^−5^ eV, a force convergence criterion for ionic steps at −10^−3^ eV/Å, and a single k-point sampling in reciprocal space. The position coordinates of the optimized structure can be found in the Appendix A. The total energies of the initial unoptimized and final optimized structures are −2966.5835 eV and −3129.5422 eV, respectively.

The optimized structure is then used to calculate the elastic coefficients ($C_{ij}$) in VASP using the stress versus strain approach of the Nielsen and Martin scheme [36]. A strain (ε) is applied to the optimized structure according to Hooks law:(1)$$\sigma_{i}=\sum_{j=1}^{6}C_{ij}\varepsilon_{j}$$
where stress component $\sigma_{i}\;$(i = 1 to 6) is linearly dependent on the applied strain $\varepsilon_{j\;}$(j  = 1 to 6) under small deformation. The stress tensor elements (xx, yy, zz, yz, zx, and xy) are used in corresponding strain. Equation (1) gives six sets of linear equations with six components of stress and 21 elastic constants. Guided by the findings from our recent work [21], the strain percentages of ±0.25%, ±0.5%, ±0.75%, ±1%, ±1.25%, ±1.5%, ±2%, ±2.25%, and ±2.5% are chosen in this work.

The elastic constants $C_{ij}$ and corresponding compliance tensor $S_{ij}$ are then used to calculate mechanical properties using Voigt’s approach, Reuss approach, and Voigt–Reuss–Hill approximation.

Voigt’s approach [37] gives the upper limit of bulk modulus $K_{Voight}$ and shear modulus $G_{Voight}$
(2)$$K_{Voigt}=\frac{1}{9}\left(C_{11}+C_{22}+C_{33}\right)+\frac{2}{9}\left(C_{12}+C_{13}+C_{23}\right)$$
(3)$$G_{Voight}=\frac{1}{15}\left(C_{11}+C_{22}+C_{33}-C_{12}-C_{13}-C_{23}\right)+\frac{1}{5}\left(C_{44}+C_{55}+C_{66}\right)$$

Reuss’s approach [38] gives the lower limit of bulk modulus $K_{Reuss}$ and shear modulus $G_{Reuss}$
(4)$$K_{Reuss}=\frac{1}{{(S}_{11}+S_{22}+S_{33})+2(S_{12}+S_{13}+S_{23})}$$
(5)$$G_{Reuss}=\frac{15}{4\left(S_{11}+S_{22}+S_{33}\right)-4\left(S_{12}+S_{13}+S_{23}\right)+3\left(S_{44}+S_{55}+S_{66}\right)}$$

Hill’s approach is the average of Voigt and Reuss approaches known as Voight–Reuss–Hill approximation (VRH) [39].
(6)$$K=\frac{K_{Voight}+K_{Reuss}}{2}$$
(7)$$G=\frac{G_{Voight}+G_{Reuss}}{2}$$
(8)$$E=\frac{9KG}{3K+G}$$
(9)$$\eta =\frac{3K-2G}{2\left(3K+G\right)}$$
where, E is Young’s modulus and η is Poisson’s ratio.

The optimized structure is then used as the input to the in house developed OLCAO (orthogonalized linear combination of atomic orbitals) package [40] for interatomic bonding calculations. The bond order (BO) values *ρ_αβ_* between any pairs of atoms are obtained from the ab initio wave functions with atomic basis expansion.
(10)$$\rho_{\alpha \beta }=\sum_{m,occ}\sum_{i,j}C_{i\alpha }^{*m}C_{j\beta }^{m}S_{i\alpha ,j\beta }$$

In the above equations, $S_{i\alpha ,j\beta }\;$are the overlap integrals between the ith orbital in the αth atom and the jth orbital in the βth atom. $C_{j\beta }^{m}$ are the eigenvector coefficients of the mth occupied molecular orbital. The BO quantifies the strength of the bond between two atoms and usually scales with the bond length (BL), but is also influenced by the surrounding atoms. The calculation of BO is based on the Mulliken scheme [41,42], hence is basis-dependent.

Total bond order density (TBOD) serves as a quantum mechanical metric utilized in material science to assess the internal cohesion of materials. TBOD is obtained by normalizing the total BO within cell volume. This work marks the first application of TBOD to biomolecules.

As is well-known, amino acids (AAs), being the basic units of proteins, are in some kind of sequential order in biomolecules. However, the interactions between them are not just with the nearest neighbor (NN) pairs (i.e., AA residues in primary sequence). In biomolecules, there are non-local (NL) interactions too. NL interaction is between AA residues that are not NN in the primary sequence but also from other nearby non-negligible bonds, which form the twists and turns in biomolecules.

We now extend our formulation and analysis of BO to amino acid–amino acid bond pair (*AABP*) [27]
(11)$$AABP\left(u,v\right)=\sum_{\alpha \epsilon u}\sum_{\beta \epsilon v}\rho_{\alpha i,\beta j}$$
where the summations are over all atoms α in AA u and all atoms β in AA v. This novel concept of *AABP* considers all bonding between two amino acid residues, including both covalent and hydrogen bonding. *AABP* is a single parameter that quantifies the interaction between two AA residues. The stronger the interaction, the larger will be the *AABP* and vice versa. *AABP* can be further resolved in NN and NL bonding.

## Figures and Tables

**Figure 1 ijms-25-10164-f001:**
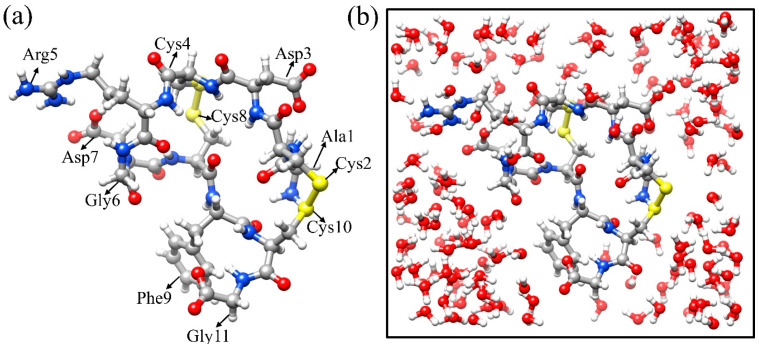
The structure of RGD (1FUV) peptide (**a**) Ball and stick figure of 1FUV with eleven amino acid residues marked. (**b**) 1FUV in a water box: blue: N, gray: C, yellow: S, red: O, and white: H.

**Figure 2 ijms-25-10164-f002:**
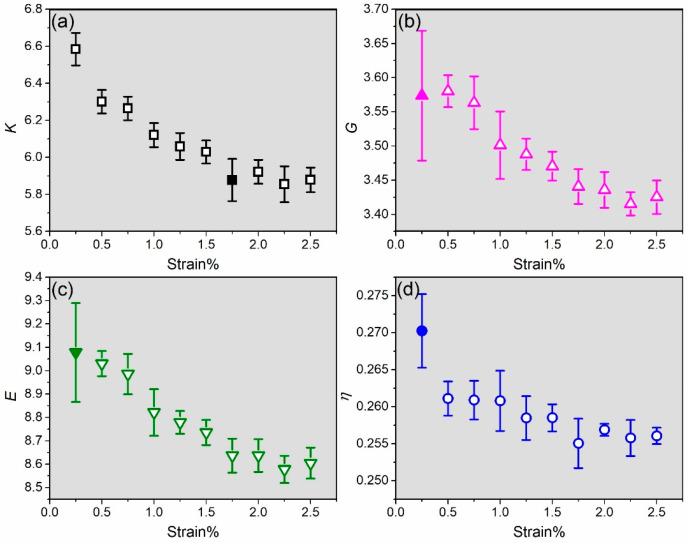
Mean value of (**a**) bulk modulus, (**b**) shear modulus, (**c**) Young’s modulus, and (**d**) Poisson’s ratio for solvated RGD (1FUV) in ten different strain percentages and calculated error bars from five sets of data. The solid symbol denotes the case with relatively high error.

**Figure 3 ijms-25-10164-f003:**
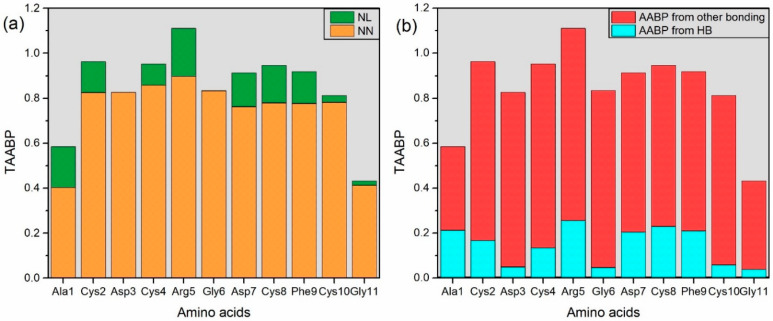
(**a**) TAABP for 11 AA residues showing contribution from non-local (NL) and nearest neighbor interactions (NN). (**b**) Contribution from hydrogen bonding and other bonding in TAABP.

**Figure 4 ijms-25-10164-f004:**
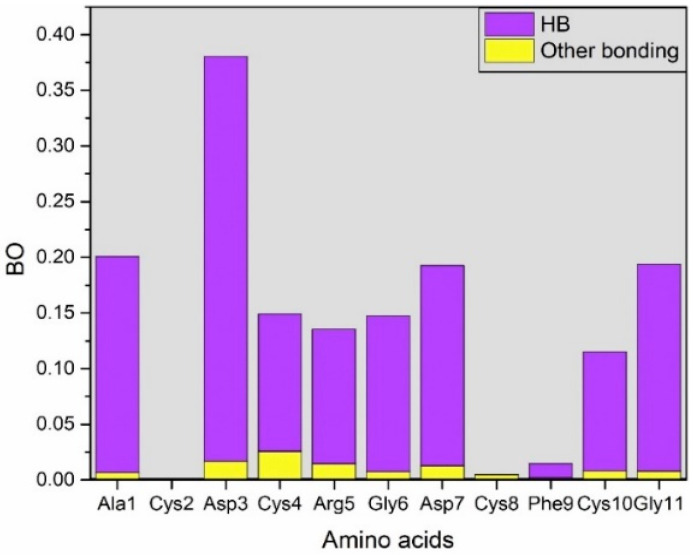
Bond order (BO) represents the interaction between amino acid residues and water molecules.

## Data Availability

The data supporting the findings of this study are available within the article and Appendix A.

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
