# Peer review of "Mechanical Properties of a Solvated Biomolecule: RGD (1FUV) Peptide"

_ijms, 2024, doi:10.3390/ijms251810164_

Round 1

Reviewer 1 Report

Comments and Suggestions for Authors

The manuscript by Adhikari et al. investigates the properties of a 1FUV polypeptide composed of Arginine, Glycine, and Aspartic acid (RGD) building units. The RGD framework is a very common tri-amino-acid sequence in biological systems known for its role in cellular. Owing to inter atomic-acid bonding, the 1FUV fonds into a polycyclic species that the authors have previously studied in detail based on computational methods (see DOI: 10.1038/srep05605). These forms of cyclic molecules, besides the obvious biological involvement, have a fundamental interest in chemistry, considering the exhibition of ring strains that have been studied in the past two centuries (starting with the Bayer strain). In this regard, the use of the Vienna Ab initio Simulation Package (VASP) for simulating the mechanical stretching of the molecule is highly appropriate as it can provide insights into the bulk modulus, shear modulus, and Young's modulus, among others.

A major issue with Adhikari et al.'s study is the ambiguity surrounding the paper's conclusions. The absence of a clear conclusion statement leaves the reader uncertain about the implications of the findings. Furthermore, while tables detailing interactions are provided, they fall short of effectively conveying the mechanical changes occurring within the polycyclic molecule, which is a complex structure not adequately addressed by linear models. Additionally, the lack of optimized structures and accompanying supporting information in the manuscript severely hampers the ability of other researchers to verify, replicate, or build upon the work presented.

My suggestion is that the authors provide their optimised models as supporting information. They should also add a conclusion statement to their article. They should also add figures showing the structural changes and calculated strains/energy relations, ideally as coloured molecular graphs.

Author Response

Dear Editor,

Thank you for your letter and detailed comments from two reviewers. Based on these comments, we have revised our manuscript accordingly by tracking changes and highlighting it by green. We have changed the title of the paper to “Mechanical properties of solvated biomolecule: RGD (1FUV) peptide” and have added one new reference. Additionally, we have also added supplementary material file. We deeply appreciate the time the reviewers spent in providing the advice with valuable comments and suggestions, which we have fully addressed. The following is our point-by-point response to these comments.

Reviewer 1:

The manuscript by Adhikari et al. investigates the properties of a 1FUV polypeptide composed of Arginine, Glycine, and Aspartic acid (RGD) building units. The RGD framework is a very common tri-amino-acid sequence in biological systems known for its role in cellular. Owing to inter atomic-acid bonding, the 1FUV fonds into a polycyclic species that the authors have previously studied in detail based on computational methods (see DOI: 10.1038/srep05605). These forms of cyclic molecules, besides the obvious biological involvement, have a fundamental interest in chemistry, considering the exhibition of ring strains that have been studied in the past two centuries (starting with the Bayer strain). In this regard, the use of the Vienna Ab initio Simulation Package (VASP) for simulating the mechanical stretching of the molecule is highly appropriate as it can provide insights into the bulk modulus, shear modulus, and Young's modulus, among others.

Response: We thank Reviewer 1 for appreciating the method we used in this study.

A major issue with Adhikari et al.'s study is the ambiguity surrounding the paper's conclusions. The absence of a clear conclusion statement leaves the reader uncertain about the implications of the findings. Furthermore, while tables detailing interactions are provided, they fall short of effectively conveying the mechanical changes occurring within the polycyclic molecule, which is a complex structure not adequately addressed by linear models. Additionally, the lack of optimized structures and accompanying supporting information in the manuscript severely hampers the ability of other researchers to verify, replicate, or build upon the work presented.

My suggestion is that the authors provide their optimised models as supporting information. They should also add a conclusion statement to their article. They should also add figures showing the structural changes and calculated strains/energy relations, ideally as coloured molecular graphs.

Response: In this work, we have conducted the first-ever calculation of the mechanical properties and total bond order density (TBOD) for a solvated biomolecule. The TBOD result suggests that the molecule exhibits reasonable cohesiveness. However, due to the absence of comparable data of similar calculations, direct comparison with other results is not possible. Given that RGD is used to target cancer cells, the calculated Poisson’s ratio and Young’s modulus are valuable quantities for furthering research in this direction. We have added following sentence in the discussion. “We would like to emphasize that this study is foundational, and it often takes years of fundamental research before such findings become applicable.”

We have provided the coordinates of the optimized structure including the cell size in the supplementary file. Additionally, we have added the following sentences in the models and methods sections. “The position coordinates of the optimized structure can be found in the supplementary materials. The total energy of the initial unoptimized and final optimized structure are -2966.5835 eV and -3129.5422 eV respectively.”

Reviewer 2:

Top of Form

Emancipation of supersoft biomolecular system, application to RGD peptide

 I would suggest a more sober and clear title. First, do not use acronyms in title.  Better use the full name: arginylglycylaspartic acid (RGD).

Even an intention of striking formulation may be acceptable, a bite-click, “emancipation” is misplaced. Note the singular “system”, namely a certain system. Which one. If correct it to systems, still there is a logical problem: one may wish to express the raising of attention devoted to “supersoft biomolecular systems”, but literally the phrase says that the molecules themselves seek emancipation.

Response: We thank Reviewer 2 for this suggestion and the title is now revised to “Mechanical properties of solvated biomolecule: RGD (1FUV) peptide”. Nevertheless, the term “RGD” is commonly used in the biological field, so we believe it is appropriate to retain the term “RGD” in the title.

The work does not deal too much with the title word “supersoft”. The word “soft” appears 3-4 times in the text, without concise focus.

Response: We have removed the word supersoft from the title. However, we would like to clarify that the word “supersoft” was used mainly to emphasize that biomaterials are relatively much softer in terms of their mechanical properties in comparison to traditional materials. 

The work is full of words-salad  constructions, such as: “Yet, a thorough comprehension of its functionality is crucial to identify the properties pivotal to the design of realms of the study.”

From the text preceding this phrase it is not clear about what functionality is the debate, but it is declared both crucial and pivotal. “Realms of the study” sounds fascinating, but is not clear what this means, let alone the puzzling turmoil about how these realms can be designed, once declared as arduous desideratum.

Response: We have edited the sentence and its following sentence to “However, to fully understand the function of biomolecules, it is essential to identify the properties—structural, dynamical, and mechanical— that are critical for design.”

Ref [1] is expected to be about X-ray crystallography on proteins but is not. E.g. if want something “historic”, look for Noble prized in 1962 or 1988.

Response: We thank Reviewer 2 for pointing out this error. Reference [1] has been replaced with: “Structural studies of protein–nucleic acid complexes: A brief overview of the selected techniques. Computational and Structural Biotechnology Journal 2023, doi:10.1016/j.csbj.2023.04.028.”

In abstract, is announced that “Herein, we present a novel computational approach that utilizes ab initio quantum mechanical method based on density functional theory.” There is no method exposed and proved as new. Besides the concatenated “computation approach” “ab initio”, “quantum mechanical” and “density functional theory” form a highly redundant construction.  Announcing “density functional theory calculations” would be sufficient.

Response: We have edited the sentence to “Herein, we present computational approach using ab initio quantum mechanical calculations to determine the mechanical properties of a solvated Arg-Gly-Asp (RGD) peptide model for the first time.” We would also like to highlight that no prior mechanical properties calculations for biomolecules using ab initio methods have been conducted, making this approach a novel technique for biomolecules.

The information is highly diluted and duplicated: the information from table 1 is duplicated in figure 2, that from table 2 in figure 3, and of course the figure 4 is nothing else than table 2. The discussion of these data is nothing else than enumerating which parameter is higher or smaller for each entry.

Response: While it is true that the data in Table 1 is also presented in Figure 2. Figure 2 includes error bars that are not shown in the Table, offering additional insights. Visual representations like figures are often necessary to complement tabular data for better understanding and clarification. Similarly, although the data from Table 2, including NL, NN, and other bonding including details on hydrogen bonding as shown in Figure 3, Table 2 also includes the number of NL AAs, which is not depicted in the figure. Regarding Table 3 and Figure 4, they do share similarities. However, we have previously received requests to present data in Tables form in addition to Figures. To further address these concerns, we have moved Table 1, Table 2, and Table 3 to the supplementary section as Tables S1, S2, and S3 respectively.

There is no reasonable info about the prepared five sets.  Essential ingredients of calculations are not given, e.g. the cell size, initial and optimized, at least the optimized one.  At the same time, because such molecule is very flexible, with many conformations, in vacuum or in water, consider also the propensity for dynamic of the water box, is incomplete and insufficient to discuss about a an optimized structure result. Besides, the presented structure look suspicions, at least in the light of the confused exposition.

Response: We thank Reviewer 2s’ demand in providing structure of the optimized model.  We have provided the coordinates of the optimized structure including the cell size in the supplementary file. We would also like to clarify that we used the same optimized structure to prepare five sets of data. However, mechanical properties on the 10 different strains were calculated five times. For each mechanical property calculation, a strain was applied in 12 different directions. Therefore, providing all 600 resulting structures (5 sets × 10 strains× 12 directions) would be extremely challenging. We are very proud of our detailed and penetrating presentation.

The authors are indicating structure taken from a PDB link given in ref [31]. But, that site gives 19 models for the given system, differing drastically in conformation.

Response: The PDB contains 19 models. And we selected the first model. We have edited the first few sentences of the Models and Methods sections as following “The structure of RGD (1FUV) is obtained from the RCSB protein data bank (PDB) [30,31], which contains 19 models with identical compositions and number of atoms. We selected the first model, which consists of 11 amino acids with a total of 135 atoms.”

The figure 1 shows the same peptide, with the exact the same conformation in both left and right side. It would be reasonable to show and discuss conformational changes for free and solvated molecules. Or between those taken from PDB and the authors results.

Response: Figure 1a and 1b show the same conformation. Figure 1a does not include any surrounding water molecule and is labeled with each AA. This figure is presented solely to illustrate the labeling of the AAs, as mentioned in the figure caption.

The presented mechanical data are not usually or presumably of interest for biological molecules and the relation with their activity. However, even if take this clue as interesting, the authors do not explain any relation with properties and with related systems, treated in this way, to set a relevance. In turn, the only comparison is carried with inorganic solids: pyrophosphate crystal, C-S-H cement mineral suolunite, amorphous-SiO2 (a-SiO2 glasses), crystalline montmorillonite clay, solvated montmorillonite clay.  The molecule is very floppy and the water box is very dynamic. The characterization by parameters of mechanical hardness is inappropriate.

Response: First, we would like to clarify that the comparison of various materials is focused on TBOD, not on the mechanical properties. As mentioned in the text, this is the first time TBOD has been calculated for a biomolecule. Additionally, calculating mechanical properties for biomolecules is uncommon, and this is probably the first time such computational project has been presented. We recognize the importance of Young’s modulus and Poisson’s ratio in cancer research, but we emphasize that this study is foundational and the beginning. It often takes years of fundamental research before such findings become applicable.

The statement “Bond order (BO) serves as a measure of bond strength” is not true.  For instance, the bond orders for C-C in CH3-CH3 and for O-O in HO-OH are both equal to 1 and yet the second molecule is more reactive, because of weaker bond.  The section about bond order does not bring valuable insight. The estimation of bond orders is routine and usually of little concern in molecular calculations (e.g. those done with Gaussian type codes). A program like VASP does not routinely offer bond orders and, then, retrieving such information may be a small challenge, but this does not make the quantities more relevant, in themselves.  At least not artificially glued to the other part of the work.

Response: We acknowledge Reviewer 2’s point; however, bond order does indicate the strength of the bond between each pair of atoms. Generally, covalent bonds are stronger than ionic bonds and thus tend to have larger bond order values. Additionally, bond order typically correlates with bond length, but since the quantum mechanical wavefunction is used to compute the bond order, the bond angle also plays a determining role. We would also like to point out that we used OLCAO method to calculate the bond order. This method has been used in a large number of our papers published in this field. Some of our relevant work, particularly in biomolecules, includes: https://doi.org/10.1039/D0CP03145C , https://doi.org/10.1021/acs.jcim.1c00560, https://doi.org/10.3390/ijms23052870 . We have also published a book titled “Electronic structure method for complex materials: The orthogonalized linear combination of atomic orbitals, Oxford, 2012”, which details all the parameters calculated by OLCAO.

With so many confusions, the material needs drastic revision.

Grammar in itself is reasonable, but there are many useless pompous phrases with low and confused information content and flawed logic.

Response: We appreciate reviewers’ feedback and have addressed all concerns point by point to the extent possible. We sincerely thank the two reviewers for their valuable input, which has helped us improve our work.

Sincerely yours,

Wai-Yim Ching,

Corresponding author on behalf of all authors.

Reviewer 2 Report

Comments and Suggestions for Authors

Emancipation of supersoft biomolecular system, application to RGD peptide

 I would suggest a more sober and clear title. First, do not use acronyms in title.  Better use the full name: arginylglycylaspartic acid (RGD).

Even an intention of striking formulation may be acceptable, a bite-click, “emancipation” is misplaced. Note the singular “system”, namely a certain system. Which one. If correct it to systems, still there is a logical problem: one may wish to express the raising of attention devoted to “supersoft biomolecular systems”, but literally the phrase says that the molecules themselves seek emancipation.

The work does not deal too much with the title word “supersoft”. The word “soft” appears 3-4 times in the text, without concise focus.

The work is full of words-salad constructions, such as : “Yet, a thorough comprehension of its functionality is crucial to identify the properties pivotal to the design of realms of the study.”

From the text preceding this phrase its not clear about what functionality is the debate, but it is declared both crucial and pivotal. “Realms of the study” sounds fascinating, but is not clear what this means, let alone the puzzling turmoil about how these realms can be designed, once declared as arduous desideratum.

Ref [1] is expected to be about X-ray crystallography on proteins, but is not. E.g. if want something “historic”, look for Noble prized in 1962 or 1988.

In abstract, is announced that “Herein, we present a novel computational approach that utilizes ab initio quantum mechanical method based on density functional theory.” There is no method exposed and proved as new. Besides the concatenated “computation approach” “ab initio”, “quantum mechanical” and “density functional theory” form a highly redundant construction.  Announcing “density functional theory calculations” would be sufficient.

The information is highly diluted and duplicated: the information from table 1 is duplicated in figure 2,  that from table 2 in figure 3, and of course the figure 4 is nothing else than table 2. The discussion of these data is nothing else than enumerating which parameter is higher or smaller for each entry.

There is no reasonable info about the prepared five sets.  Essential ingredients of calculations are not given, e.g. the cell size, initial and optimized, at least the optimized one.  At the same time, because such molecule is very flexible, with many conformations, in vacuum or in water, consider also the propensity for dynamic of the water box, is incomplete and insufficient to discuss about a an optimized structure result. Besides, the presented structure look suspicions, at least in the light of the confused exposition.

The authors are indicating structure taken from a PDB link given in ref [31]. But, that site gives 19 models for the given system, differing drastically in conformation.

The figure 1 shows the same peptide, with the exact the same conformation in both left and right side. It would be reasonable to show and discuss conformational changes for free and solvated molecules. Or between those taken from PDB and the authors results.

The presented mechanical data are not usually or presumably of interest for biological molecules and the relation with their activity. However, even if take this clue as interesting, the authors do not explain any relation with properties and with related systems, treated in this way, to set a relevance. In turn, the only comparison is carried with inorganic solids: pyrophosphate crystal, C-S-H cement mineral suolunite, amorphous-SiO2 (a-SiO2 glasses), crystalline montmorillonite clay, solvated montmorillonite clay.  The molecule is very floppy and the water box is very dynamic. The characterization by parameters of mechanical hardness is inappropriate.

The statement “Bond order (BO) serves as a measure of bond strength”  is not true.  For instance, the bond orders for C-C in CH3-CH3 and for O-O in HO-OH are both equal to 1 and yet the second molecule is more reactive, because of weaker bond.  The section about bond order does not bring valuable insight. The estimation of bond orders is routine and usually of little concern in molecular calculations (e.g. those done with Gaussian type codes). A program like VASP does not routinely offer bond orders and, then, retrieving such information may be a small challenge, but this does not make the quantities more relevant, in themselves.  At least not artificially glued to the other part of the work.

With so many confusions, the material needs drastic revision.

Comments on the Quality of English Language

Grammar in itself is reasonable, but there are many useless pompous phrases with low and confused information content and flawed logic.

Author Response

(The authors gave the same response as above.)

Round 2

Reviewer 1 Report

Comments and Suggestions for Authors

The authors have responded to my main concern  I understand that the authors have included a discussion section, but including a conclusion statement will complete the article.

Author Response

The authors have responded to my main concern I understand that the authors have included a discussion section, but including a conclusion statement will complete the article.

Response: We have added a conclusion section as following

“In this study, we have, for the first time, calculated mechanical properties of a solvated RGD peptide using ab initio quantum mechanical calculations. Additionally, we performed bonding analysis, including AABP and TBOD, highlighting RGD’s cohesion and its relevance in cancer research due to its affinity for integrin. Our future work will extend this approach to other biomolecules, aiming to develop predictive models based on amino acid residue composition.”